# Healthcare Associated Infections: An Interoperable Infrastructure for Multidrug Resistant Organism Surveillance

**DOI:** 10.3390/ijerph17020465

**Published:** 2020-01-10

**Authors:** Roberta Gazzarata, Maria Eugenia Monteverde, Carmelina Ruggiero, Norbert Maggi, Dalia Palmieri, Giustino Parruti, Mauro Giacomini

**Affiliations:** 1Healthropy S.r.l., 17100 Savona, ItalyCarmelina.Ruggiero@dibris.unige.it (C.R.); norbert@ieee.org (N.M.); Mauro.Giacomini@dibris.unige.it (M.G.); 2Department of Informatics, Bioengineering, Robotics and Systems Engineering (DIBRIS), University of Genova, 16145 Genoa, Italy; 3Epidemiology Office, Azienda Unità Sanitaria Locale (AUSL) di Pescara, 65124 Pescara, Italy; dalia.palmieri@gmail.com; 4Department of Infectious Disease, Azienda Sanitaria Locale (AUSL) di Pescara, 65124 Pescara, Italy; parruti@tin.it

**Keywords:** MDROs, interoperability, semantic harmonization, syntactic harmonization, data extraction, surveillance systems

## Abstract

Prevention and surveillance of healthcare associated infections caused by multidrug resistant organisms (MDROs) has been given increasing attention in recent years and is nowadays a major priority for health care systems. The creation of automated regional, national and international surveillance networks plays a key role in this respect. A surveillance system has been designed for the Abruzzo region in Italy, focusing on the monitoring of the MDROs prevalence in patients, on the appropriateness of antibiotic prescription in hospitalized patients and on foreseeable interactions with other networks at national and international level. The system has been designed according to the Service Oriented Architecture (SOA) principles, and Healthcare Service Specification (HSSP) standards and Clinical Document Architecture Release 2 (CDAR2) have been adopted. A description is given with special reference to implementation state, specific design and implementation choices and next foreseeable steps. The first release will be delivered at the Complex Operating Unit of Infectious Diseases of the Local Health Authority of Pescara (Italy).

## 1. Introduction

This paper is an extension of work originally presented in pHealth 2019 [1]. Multi drug resistant organisms (MDROs) infections are responsible for an increasing number of deaths each year. The correct estimation of the number of such infections is difficult because of the lack of a specific International Classification of Disease (ICD) code for this type of infection: the ICD is the global standard for international comparability in the collection, processing, classifying and reporting of mortality statistics. With over-invasive use of antibiotics, the establishment of centres for reproduction and transmission of MDROs (hospitals for acute long-term and nursing facilities), and increased rates of iatrogenic immunosuppression, the population at risk of MDROs infections and the probability of drug resistance will continue to increase [2]. It has been estimated that 700,000 deaths worldwide are caused by antibiotic-resistant bacterial infections and that the number could increase to 10 million per year in 2050 [3]. Although estimation methods have been widely criticised [4,5] and new approaches to the antimicrobial resistance (AMR) death system need to be considered [6], it is clear that controlling antibiotic resistance is a global public health concern.

Several studies on germs show the importance of the problem [7,8,9]. In this respect, the development of surveillance systems for measuring and comparing the frequency of bacterial resistance is essential for the evaluation of MDROs infections [10]. Better surveillance and reporting mechanisms need to be created [2].

In recent years attention to the prevention and surveillance of healthcare associated infections (HAIs) has increased to a great extent, due to an increased awareness of patient safety and potential harm prevention [11,12]. The World Health Organization (WHO) and the European Centre for Disease Prevention and Control (ECDC) have set up networks called GLASS (Global Antimicrobial Resistance Surveillance System in 2015) [13] and EARS-Net (European Antimicrobial Resistance Surveillance Network in 1998) [14] respectively in order to collect worldwide and European pseudo-anonymised microbiological data on the eight most relevant pathogens for human infections (*Escherichia coli*, *Klebsiella pneumoniae*, *Pseudomonas aeruginosa*, *Acinetobacter baumannii* group, *Staphylococcus aureus*, *Streptococcus pneumoniae*, *Enterococcus faecalis*, *Enterococcus faecium*). They are isolated from certain samples (blood, urine, faeces, urethral swab and cervical swabs (GLASS) and blood and cerebrospinal fluid (EAST-NET)). To date 85 countries are participating in GLASS and 30 European Union (EU) and European Economic Area (EEA) countries are participating in EARS-Net. In order to participate in these networks, each country must create a specific national network to collect microbiological data produced by microbiological laboratories, harmonize them, analyse and process them and send them using specific formats both syntactic and semantic. 

An example of a national network is the Surveillance System AR-ISS (Antibiotic Resistance-Health Superior Institute) established by the Italian Ministry of Health in 2001 with the aim of collecting microbiological data only from laboratories that comply with specific recruitment criteria in the execution of tests, harmonize them manually and process them [15]. AR-ISS requires all participating laboratories to provide the results of antibiograms carried out only on samples selected by the GLASS and EAST networks on a yearly basis. This only relates to eight pathogens and to a small group of antibiotics for each pathogen. No format standard (usually excel files) nor semantic standards (local terminologies are used) are adopted. Therefore, before the data are processed, a manual interpretation of all data has to be carried out and then the data are harmonized adopting common coding. This results in a strong limitation that could be overcome by the adoption of standardized procedures and of information and communication technology (ICT) based data processing. The set of the antibiograms is limited by the manual procedure which is at present adopted.

The data provided by surveillance networks are relevant for trend monitoring even though they do not give a complete epidemiological view, specifically for measuring the effect of action plans. Using data from EARS-Net, Cassini et al. estimated the influence of infections from 16 antibiotic resistant bacteria. The results of this study showed that the burden of MDROs infections is greater in infants and elderly people and has increased since 2007. Specifically, Italy and Greece have the highest number of estimated burdens of antibiotic resistant bacteria infections compared to the rest of other EU and EEA countries [16].

Summarizing, considerable surveillance of HAIs is required, however manual surveillance methods are labour intensive and lack standardisation. In this respect a standard based approach is extremely important. The work described here is focused on setting up an interoperable infrastructure that can be adopted by different laboratories in which different methods and procedures are used. Information technology based systems have the potential to improve HAIs to a great extent [12,17], enhancing reliability, efficiency and surveillance standardization [18]. Moreover, the adoption of electronic surveillance software brings about considerable improvements as relates to data collection and case ascertainment while maintaining high levels of sensitivity and specificity [18]. The use of electronic health records (EHRs) and electronic medical records (EMRs) has been found to provide opportunities to improve healthcare associated infections surveillance [19], since traditional manual surveillance methods are limited by inter-observer variability and labour intensive [20], whereas electronic surveillance systems have been found to be accurate and potentially time saving [21,22,23].

An interoperable infrastructure for MDROs surveillance has been developed, for the Abruzzo Region in Italy, with a view to extend it to other regions. The system described is based on EHRs and EMRs, which have become available in many hospitals over the last decades and on an ancillary database. Data protection, privacy policy and standards have been adopted according to the EU General Data Protection Regulation (GDPR).

## 2. Materials and Methods 

In order to achieve the goals in the Regional Prevention Plan (RPP) and allow the Abruzzo region to actively participate in various national and international surveillance systems, it has been decided to design and implement an infrastructure which supports two independent systems which communicate with each other:a monitoring and notification system for MDRO;a monitoring system for hospital and drug prescriptions landscape related to infections and microbiological data.

All data are automatically stored in a common repository hosted within Abruzzo Region infrastructures. This solution has been adopted because the Italian public administration does not allow to use a cloud-based data infrastructure due to concerns about privacy. The data include pseudo-anonymized data from microbiology reports, administration data (for hospitalized patients), discharge summarization notes and drug prescriptions (for non-hospitalized patients) produced by the Hospital Information System (HIS) of by the four districts of the Abruzzo region: Pescara, Chieti, Teramo and L’Aquila (Figure 1). The data are not wholly anonymized in order to acquire the information about all prescriptions in the last six months for each patient and to monitor for possible recurrences. All collected data have then been processed and used with different purposes, such as for example:monitoring germs MDROs and related infections, notification of infective diseases and adverse reaction to vaccines, monitoring antibiotic usage in hospital and landscape;implementing support actions for antibiotic prescriptions, monitoring/early TBC diagnosis, screening/early diagnosis of HIV.

The following sections describe the design strategy of the overall system, the collected data, the medical informatics standards for semantic and syntactic harmonization and the methods for connection with HIS.

### 2.1. Overall System Design

The overall infrastructure has been planned adopting the service oriented architecture (SOA) paradigm, because it allows designing distributed solutions in which existing and new applications can be aligned and integrated [24,25,26,27,28,29]. This approach was also adopted in the same context for a web-based multidrug-resistant organisms surveillance and outbreak detection system [30]. Tseng et al. combined the SOA with HL7 (Health Level Seven) standards. This solution has been adopted in order to guarantee the first two interoperability levels that is technical and semantical interoperability [31]. The third and last level, the process interoperability, has been supported adopting the standards by the HL7 and OMG (Object Management Group) program Healthcare Services Specification Project (HSSP) [31,32]. The HSSP aim is to provide functional and technical specifications for the design and the implementation of web services interfaces, therefore it has been decided to adhere to some of these specifications combining and modifying when necessary some web services, compliant to specific HSSP standards, already designed and developed.

The system has three main components: the Health Record Management Service (HRMS), the Health Terminology Server (HTS) and the Patient Identity Service (PIS). HRMS has been the first component that has been implemented following the HSSP specifications [24]. It is responsible for the management of patients’ reports, clinical history and profiles and its interfaces are compliant with the Retrieve, Locate and Update Services (RLUS) Release 1 standard [29,33,34]. The HRMS was already adopted in different solutions to monitor patients affected by Congestive Heart Failure (CHF) [24] and to share and re-use clinical data in ophthalmology [35] and in infectiology [31,36]. The second component was the Health Terminology Service (HTS). Its interfaces are compliant to the Common Terminology Service Release 2 (CTS2) [37,38] and its aim is to correctly manage semantics—a fundamental aspect when it is necessary to compare clinical data from different care facilities which adopt specific local terminologies [39]. Finally, some of the authors designed and developed the Patient Identity Service (PIS), which is a solution to manage identifiers, defined within different care facilities, and the relations among them—a crucial topic to manage the patient transfers from one hospital to another [31]. The PIS interfaces are compliant to the Identification and Cross-Reference Service (IXS) Release 1 [40,41]. As described in the Results section, one of the stored identifiers within the PIS will be the Italian National Insurance Code (fiscal code), the national unique patient identifier. It is formed by 16 character and is obtained with an algorithm that use some sensitive patient’s data (e.g., the name, the birth date etc.), therefore it must be encrypted to guaranty patient’s privacy. For this reason, the Advanced Encryption Standard (AES) has been adopted because a symmetric cryptography is useful [42].

### 2.2. Data Collection

In order to store all pseudo-anonymized data, it has been necessary to introduce a Central Repository in the overall architecture. A relational database has been created and the Entity-Relation diagram has been designed starting from the analysis of the documents provided by the national network AR-ISS [43] and by the international networks EARS-Net [44] and GLASS-Net [45], which describe all data required by each network. In this way, the system can provide all data to these networks. In addition, information not managed by the networks, such as drug prescription and discharge summarization notes, has also been included. At the end of the requirement analysis, the patient’s data to be managed are: gender, birth year, possible death date, weight, height, nationality, residence district. For the monitoring and notification system for MDROs germs and tuberculosis, the further data relating to microbiological reports and discharge summarization notes have also been considered. Specifically, the data from all laboratory reports are administrative information (e.g., entry order identifier, entry order date, close date, validation date, care facility etc.), specimen, culture and PCR (Polymerase Chain Reaction) text (e.g., searched organism, result, eventual note etc.), isolated organism (e.g., name, code, quantization, note, alert organism etc.) and its antibiogram (e.g., antibiotic, value, sensitivity, methodology (MIC (Minimal Inhibitory Concentration), AST (Antimicrobial Susceptibility Testing) etc.) [46]. While the national and international networks consider only a very restricted set of specimens, pathogens and antibiotics, it has been decided to consider all specimens, pathogens and antibiotics present in all microbiological reports by the public laboratories of the Abruzzo region. The data from the discharge summarization notes relating to admission, possible transfers and discharge of all patients for which at least one culture or PCR text result was positive in the last six months. Moreover, the information from the hospital monitoring system and landscape drug prescription for infections and microbiological data have been associated to the antibiograms in the microbiological reports and to all drug prescriptions (e.g., drug name and code, dose, frequency) and administrations (e.g., drug name and code, dose, datetime etc.) for the same set of patients in the discharge summarization notes.

In order to guarantee privacy, all patient data have been pseudo-anonymized by an application hosted within the specific HIS, as described in the next sections, adopting an identifier defined by the Central Repository, called ID_Patient_Central_DB. 

### 2.3. Semantic and Syntactic Harmonization

In order to compare clinical and administrative data from different care facilities, a harmonization process is needed. The semantic harmonization process adopted here is similar to the one described in the terminology system within an Italian Electronic Health Record [39]. From a semantic point of view, national and international vocabularies have been adopted for coding the following information:ISTAT (Italian National Statistical Institute) classification for the territorial administrative units [47] for patient’s nationality and residence district;NSIS (National New Health Information System) local health authority and hospitalization facilities [48] for the involved care facilities;HL7 Specimen Type [49] for the analysed specimen;HL7 Observation Interpretation [50] for the antibiogram sensitivity;LOINC (Logical Observation Identifiers Names and Codes) vocabulary [51] for the culture and PCR tests and patient’s weight and height;NHSN (National Healthcare Safety Network) organism list [52];Therapeutic Chemical (ATC) Classification System [53] for antibiogram antibiotics and drug prescriptions;AIFA (Drug Italian Agency) AIC (Marketing Authorization) [54] for drug administrations;ICD-9 (International Classification of Diseases, Ninth Revision) [55] for admission and discharge diagnosis.

For syntactic harmonization, different HL7 products and a national standard have been adopted:HL7 v3 CDA R2 (Clinical Document Architecture Release 2) [56] for microbiological reports and drug prescriptions and administrations;HL7 FHIR (Fast Healthcare Interoperability Resources) v4.0.0 Patient [57] with the HL7 extension for the nationality [58] for patient’s administrative data;HL7 FHIR v4.0.0 Observation [59] for patient’s weight and height;Italian Health Ministry Tracing B—Information Hospitalization of the SDO (Hospital Discharge Form) [60] for discharge summary notes.

### 2.4. Connection with the HIS

In order to automatically extract pseudo-anonymized data from the HIS of the involved care facilities, specific client applications have been designed and developed. The HIS application queried by this type of clients are the LIS (Laboratory Information System), the PAS (Patient Administration System), the DDS (Drug Distribution System) and the EMR and EHR. The technical aspects of each client depend on two aspects: on the one hand, on the data storage modality in each database and, on the other hand, on the data extraction strategy from the application with which vendors provide the authors. Initially, the activity has been focused on the Pescara Local Health Authority (LHA) and its connection to each application of the HIS. The LIS of Pescara LHA (Modulab) has been provided by a software company (Werfen, Milan, Italy) and it is not possible, at present, to query its database via ODBC (Open Data Base Connectivity). Therefore, the data extraction from Modulab is semi-automatic, and takes place by manual uploading of spreadsheet files which are manually extracted through a Modulab tool. The “Areas ADT-web” program, provided by the Engineering Ingegneria Informatica company (Rome, Italy), is used as PAS in Pescara and the connection with its database is direct and automatic through query of views provided via ODBC. In order to get patient’s administrative information related to each microbiological report, the Engineering company provided a specific view that, starting from the entry order identifier, returns gender, birth year, nationality, residence district and local patient identifier (Id_patient_HIS_Pescara). Moreover, the vendor created another view that, starting from the Id_patient_HIS_Pescara and a date, returns all hospitalization information from the selected date to the moment of the query. The “Thema” program provided by the Kiranet Company (Aversa, Italy), is used as DDS in Pescara and the data extraction is automatic through query via ODBC. Kiranet company developed a similar view that, starting from the Id_patient_HIS_Pescara and a date, returns all drug administrations from that date to the moment of the query. The Pescara LHA has no EMR, therefore, at present, the information about weight and height cannot be sent to the Central Repository. A client has been set up to manage the observations when the electronic data will be available. 

## 3. Results

The infrastructure for regional monitoring and notification of MDROs germs and the regional monitoring system for hospital and landscape drug prescriptions related to infections and microbiological data is shown in Figure 2. The architecture is formed by the following components:A central database (Central Repository in the figure) that stores all pseudo-anonymized data to monitor;Four client applications (Extraction Clients), hosted within the HIS of each involved LHA, responsible for the data extraction and pseudo-anonymization;A client application (Monitoring and Notification Client) that monitors pathogens and drug prescriptions;A client application to extract data from the Central Repository and send information to the external national and international networks (Dispatch Client);An orchestrator based on a set of algorithms and web services providing functionalities that allow:The four Extraction Clients to harmonize and send pseudo-anonymized data to the Central Repository;Monitoring and Notification Client to generate emails reporting processing (in aggregate form) of data stored in the Central Repository;Dispatch Client to extract (in analytical form) and send processed and harmonized data, stored in the Central Repository, adopting the specific required terminologies and formats for each network.A PIS to manage patient identifiers.

In the next sections, details about each component and its implementation state are provided.

### 3.1. Central Repository

A Central Repository has been set up in which all pseudo-anonymized data previously described are stored, referring to the ID_Patient_Central_DB—a fake and incremental identifier defined by this database. 

The repository is a SQL Server database obtained from Microsoft (Redmond, WA, USA) hosted on a server (Intel Quad Core, 16 GB RAM (Random Access Memory), 1 TB HDD (Hard Disk Drive), 64-bit Windows Server 2016 Standard Edition) installed within the Abruzzo regional information systems (Server 1 in the Figure 2). Direct access to the database, through Microsoft SQL Server Management Studio (SSMS), is available only to the authors, that can perform extraction queries to obtain aggregated data for statistical and research proposes. The client applications can access data only by the functionalities provided by the interfaces of the orchestrator, according to the following access policy:Writing rights of all administrative and clinical data for the four Extraction Clients;Reading rights of all administrative and clinical data for the Monitoring and Notification Client and the Dispatch Client;Reading rights of the ID_Patient_Central_DB for patients to be monitored (that is the ones for which at least one culture or PCR text result was positive in the last six months) for the four Extraction Clients.

### 3.2. Orchestrator

This component is the core of the overall architecture. It allows getting and putting harmonized data from and into the Central Repository, coordinating the communication between the involved applications and services. It is formed by a HRMS and an HTS, designed and implemented starting from the web services which were setup in previous research work [24,31]—a set of algorithms satisfying the project requirements and a Central Repository Identity Service (CRIS). The CRIS is a web service designed and implemented starting from the previously developed PIS which provides the subset of IXS functionalities for patient entity management within the Central Repository.

The web service interfaces are compliant with the HSSP standard and use as Semantic Signifiers (that is “the manifestation of a computable information model, tagged with a name and version and capable of being used and enforced by reference” [32]) the HL7 products and the national standards mentioned in the Materials and Methods section. The orchestrator has been designed to be hosted on the same server on which the Central Repository is installed (Server 1), and the communication between these solutions takes place through the execution of stored procedures called via ODBC. From a technical point of view, the orchestrator is formed by a set of Windows Communication Foundation (WCF) services whose interfaces are only visible by the authorized clients, which have been installed within the Abruzzo region Virtual Private Network (VPN). The operations provided to the client applications through the interfaces, compliant with the RLUS and the IXS standards, can be divided into two categories: writing functionalities and reading functionalities.

The first set of operations, which can be called only by the four Extraction Clients, are:From the CRIS IXSManagementAndQueryInterface:
CreateIdentityFromEntity: given gender, birth year, nationality and residence district (wrapped in a HL7 FHIR v4.0.0 Patient object), it creates a new ID_Patient_Central_DB;UpdateEntityTraitValues: given the ID_Patient_Central_DB, it allows to update the nationality and the residence district of the specific patient (wrappe in a HL7 FHIR v4.0.0 Patient object).From the HRMS RLUSManagementAndQueryInterface:
Put: it allows to semantically and syntactically harmonize and send to the Central Repository patient’s clinical information wrapped in either HL7 v3 CDA R2 objects (microbiological reports and drug prescriptions and administrations), or Tracing B—Information Hospitalization of the SDO object (discharge summarization notes) or HL7 FHIR v4.0.0 Observation resources (weight and height).

The reading functionalities provide access to administrative and clinical pseudo-anonymized data in analytical or aggregate form. The standard methods, assembled according to the client application authorized to call them, and interfaces are shown below.
Monitoring and Notification Client: from the HRMS RLUSManagementAndQueryInterface:
Get: given the parameters required to monitor in aggregate form, it allows to generate the text of the newsletter in XML (eXtensible Markup Language) format.Dispatch Client: from HRMS the RLUSManagementAndQueryInterface:
List: given the parameters required to the specific network, it generates an XML file containing the list of all analytical data to be sent to the network, adopting the specific terminology and formalism.Four Extraction Clients: from the CRIS IXSManagementAndQueryInterface:
FindIdentitiesByTraits: it allows to get the list of all patients (ID_Patient_Central_DB) which must be monitored at the moment.

At present the first release of the orchestrator is being tested.

### 3.3. Patient Identity Service

In order to guarantee patient’s privacy, all data deriving from the HIS are pseudo-anonymized according to the patient identifier defined within the Central Repository—that is the mentioned ID_Patient_Central_DB. In order to manage the monitoring of the landscape drug prescriptions and possible patient transfers from and to different regional care facilities, a PIS has been included in the architecture. This service stores and makes available—to the Extraction Clients only—the map between the encrypted National Insurance Code and the ID_Patient_Central_DB, the nationality and the residence district of each patient. Only Extraction Clients know the encryption key, therefore only these applications are able to decrypt the scrambled National Insurance Code to get the original identifier. The PIS is hosted within the Abruzzo regional information systems as the Central Repository and the orchestrator on a different server with the same technical features (Server 2 in Figure 2). The reason for this choice is to reduce the risk of tracing the identity of the owner of the data stored in the Central Repository.

As shown in Figure 2, the PIS is formed by a Microsoft SQL Server database, which stores the encrypted Fiscal Code, the ID_Patient_Central_DB, the nationality and the residence district of the patients, and two WCF services. Again, the content of the PIS database is available only to the four Extraction Clients through the WCF services that communicate with the database through the execution of stored procedures called via ODBC. These two WCF service interfaces are compliant with IXS standard and use the HL7 FHIR v4.0.0 Patient as Semantic Signifier. Specifically, the first service represents the IXSManagementAndQueryInterface for which the following functionalities have been implemented:RegisterEntityWithIdentity: it allows to register the patient’s identifiers, that is the scrambled Fiscal Code and the ID_Patient_Central_DB, the nationality and the residence district;ListLinkedIdentities: given the scrambled Fiscal Code, it returns, if it exists, the corresponding ID_Patient_Central_DB, the nationality and the residence district;UpdateEntityTraitValues: given the ID_Patient_Central_DB, it allows to update the nationality and/or the residence district of the specific patient.

The second service implements the IXSAdminEditorInterface, with the following method:LinkEntities: it allows to store the association between the encrypted Fiscal Code and the ID_Patient_Central_DB in the repository.

At present, this service is ready to be hosted on a dedicated server.

### 3.4. Extraction Clients

It is planned to install a specific type of components to extract the data to feed the two monitor systems. The Extraction Clients will be hosted on a server whose technical features are the same as the previous ones. Specifically, it will be based on:A client application that intercepts, pseudo-anonymizes and sends to the Central Repository, through calls to the orchestrator, data stored within the databases of the specific LIS, PAS, DDS and EMR;A very simple Microsoft SQL Server database to store the data necessary to the correct extraction (e.g., the patient’s encrypted Fiscal Code, the association of the LHA patient identifier ant the ID_Patient_Central_DB and the lost date of data extraction from each HIS application);A web application to visualize—given the ID_Patient_Central_DB—the specific LHA patient identifier (e.g., Id_Patient_HIS_Pescara in Figure 3) this application can be used by authorized staff only.

The communication between the database and the client application or the web application takes place through the execution of stored procedures called via ODBC. In addition, the client application also interacts with the orchestrator and the PIS. Specifically, whenever a new patient is extracted the PIS is queried to check if a corresponding ID_Patient_Central_DB is present. If this is the case the association between the LHA patient identifier and the ID_Patient_Central_DB is stored in the database and the identifier is used to pseudo-anonymize all the data before sending them to the orchestrator. If not, a new ID_Patient_Central_DB is obtained from the CRIS and the new association is stores both in the database and in the PIS.

In this first year of activity, the design and the implementation of the Extraction Client of the Pescara LHA has been carried out. A general extraction algorithm has been developed and that is ready to be tested. The data extraction is semi-automatic, therefore this specific client application is a web application that allows authorized staff to manually upload the spreadsheet files which contain all microbiological reports monthly produced by the LIS. This uploading, for the HIS of the Pescara LHA, is the trigger event of the data extraction.

### 3.5. Monitoring and Notification Client and Dispatch Client

In order to generate and send monitoring and notification mails, a client has been designed and implemented, which will be hosted on the same server as for the orchestrator and the Central Repository (Server 1). This client is formed by a web application to generate newsletters, and by a repository to store email addresses. This allows to set the relevant parameters used by the orchestrator to extract the data in aggregate format and to generate the text of the newsletters. In order to obtain the contacts of the people to whom the newsletters have to be sent, this application communicates with the Repository.

Another client application will be hosted on Server 1 to extract (in analytical format) all data required by the national AR-ISS Net and the international GLASS-Net and EAST-Net, adopting the specific terminologies and formats. At present, both the Monitoring and Notification Client and Dispatch Client are at prototypal design level.

## 4. Discussion 

A standardized prototype was developed, in the past, for one laboratory for sharing antibiotic resistance data according to the HL7 standard [61]. The present paper describes its extension to many laboratories with a view to take in to account Italian and international observational studies. Moreover, a structure for data comparison among laboratories for the management of MRDOs involving Italy and France within a cross-border EU cooperation program was implemented. This resulted in significant improvements in the treatment of infection diseases according to international guidelines [62].

The standardized system described here extends these works achieving one interoperable infrastructure which can be used by different laboratories setting up one harmonization layer which converts local terminologies and structures into the terminology and structure of the standardized system.

This infrastructure supports two surveillance systems to monitor MDROs infections and hospital and landscape drug prescriptions and administrations. It is formed by a Central Repository, an orchestrator, a PIS and a set of client applications. 

The Central Repository is a relational database that stores all data to be monitored. Its access is provided to authorized clients through the orchestrator service interfaces, which are compliant to specific HSSP project standards, which allow interoperability at technical, semantic and process levels.

The orchestrator is also responsible for data harmonization—a fundamental aspect for process information by different care facilities, and the semantic interpretation—a crucial point to obtain the correct complete clinical view.

The PIS was introduced to correctly manage the association between the different patient’s identifiers and to support the data pseudo-anonymization process, realized with the adoption of the patient identifier defined within the Central Repository, the mentioned ID_Patient_Central_DB.

The client applications can be basically divided into two categories: applications that put data in the Central Repository and applications that get data from the Central Repository. The first group is formed by the four Extraction Clients, each hosted within a HIS of the involved LHA, that intercept the required data from the specific LIS, PAS, DDS and EMR and send them to the Central Repository. The second category includes the Monitoring and Notification Client, which generates newsletters reporting aggregate elaborations of data stored in the Central Repository, and the Dispatch Client, which extracts and harmonize data from the Central Repository analytical to send them to the Italian Health Ministry, in order to participate to national and international surveillance networks.

All specimens, pathogens and antibiotics present in all microbiological reports produced by public laboratories of the Abruzzo region are collected in the Central Repository. In this way a complete view of both presence and state of antibiotic resistance of the pathogens present in the region is available enabling early prevention of the diffusion of organisms that would originate critical situations. Initially only antibiotic prescriptions and administrations were considered, but at a later stage possible inhibition of the antibiotic effect on organisms and possible activation of resistance mechanism due to other drugs have been considered. For this reason, it is fundamental to monitor the landscape administration too. In the proposed architecture, the authors did not write about this aspect because they still have to establish how to extract these data. At present, the strategy that seems to be the better one is to interact with the repository of the national IT company, controlled by the Italian Economy and Finance Ministry, responsible for monitoring the population health cost. This is based on the fact that Italians can partially subtract from taxes expenses for drug purchases by Fiscal Code specification. Querying this repository would allow to monitor most drugs which are purchased by patients. This would require authorization by the Regional Data Protection Officer. In this context the consent of the regional ethics committee it is fundamental. The authors have proposed a solution to be hosted on physically separated servers to reduce the risk of tracing the identity of the owner of the data stored in the Central Repository. At present, the authors are going to show this committee the overall architecture, in order to prove that this risk is very low and can be hazarded, since community health protection is more important than individual’s privacy protection.

The complete authorization procedure is expected to be completed in a few months, so the components will be installed on the dedicated servers and start data processing to obtain the first results, according to the deadlines indicated by the National Prevention Plan (NPP) and the RPP. These results will be essential to decide the content of the newsletters and to complete the design and development of the Monitoring and Notification Client. Moreover, they could be used to provide the Italian Health Ministry with a new set of possible pathogens and specimens to be considered in the AR-ISS Net. These could also be considered in the EAST-Net and GLASS-Net. As relates to the latter networks Italy is not participating in them for several reasons. Manual harmonization of the data collected from all national laboratories, needed before the required data processing, is one of the most relevant ones. The solution presented here allows to collect data from all regional laboratories and to extract them adopting the specific semantic and syntactic formalism, and could therefore be a good starting point to facilitate collection and processing at national level.

In the Results section, the Extraction Client is described indicating that it has several components, among which a web application to visualize the specific LHA patient identifier, given the ID_Patient_Central_DB. In order to participate in the AR-ISS net the Italian Health Ministry requires to trace the identity of an ID_Patient_Central_DB. The regional representative may be asked further information on a specific patient in order to focus on a specific clinical case. In this respect, the Extraction Client for the HIS of the Pescara LHA indicates the methods to obtain data from each HIS application database. At present, the data extraction is semi-automatic because a direct access to the LIS database is not possible. In order to manage this and possible changes of providers, it is intended to ask the HIS administrators to require tender procedures with direct data access, ether by extraction procedures already being used or by international and national standards.

## 5. Conclusions

Semantic analyses of reports have shown that the most relevant data are not the analytical ones provided by the antibiogram, but rather the ones in the note-text section. Therefore, a Natural Language Processing (NLP) algorithm is being setup to interpret the notes in the reports, in order to extract and store additional semantically enriched data in the Central Repository.

At present, the overall solution is under evaluation by the regional ethics committee, to certify that this risk to trace the identity of the real owner of the data stored in the Central Repository is very low and can be hazarded, because protecting community health is more important than protecting the privacy of the individual. This step is fundamental in order to obtain the regional DPO authorization to install the devolved components and to start to adopt the system in clinical practice in order to achieve the first results, according to the deadlines indicated by the NPP and the RPP.

Similar systems have been set up and are being used in another region in Italy for a clinical network for human immunodeficiency virus [63] and for ophthalmology [35,64].

Infectious disease management is going to be greatly improved by the introduction of genetics information. The integration of next generation sequencing with clinical data from EMRs is regarded as a key aspect that will transform infectious disease management [65] and support the development of precision-based approaches for the treatment of drug-resistant and bacterial infections [66]. It is intended to include genomic data in the system integrating genotype and phenotype information in the EMR and EHR. This integration has been given increasing attention, since genome sequencing is being more broadly adopted in medical practice [67,68,69,70]. Several diseases have been recently considered in this respect [71,72,73,74,75,76].

## Figures and Tables

**Figure 1 ijerph-17-00465-f001:**
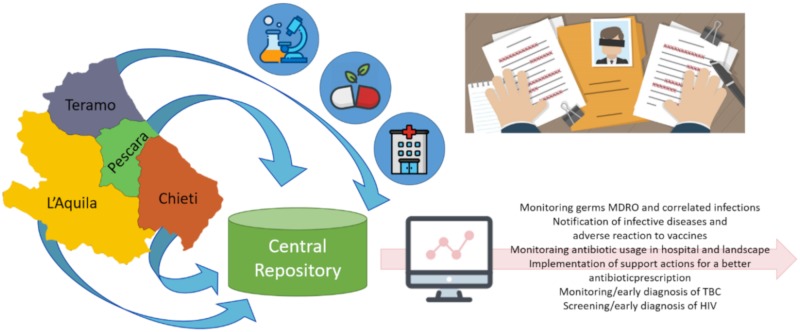
Microbiology data, administration data, discharge notes and drug prescriptions from four provinces are collected into the Central Repository of the infrastructure.

**Figure 2 ijerph-17-00465-f002:**
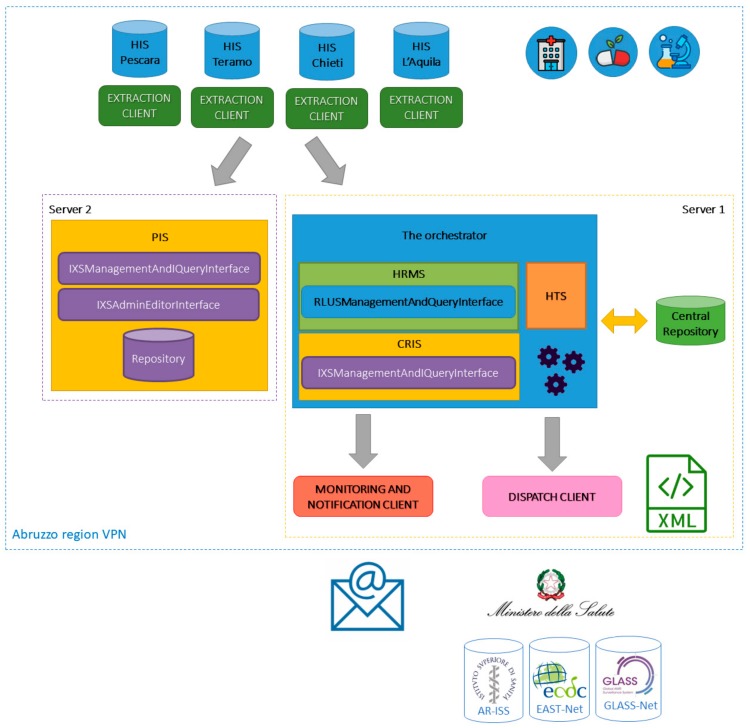
The overall architecture.

**Figure 3 ijerph-17-00465-f003:**
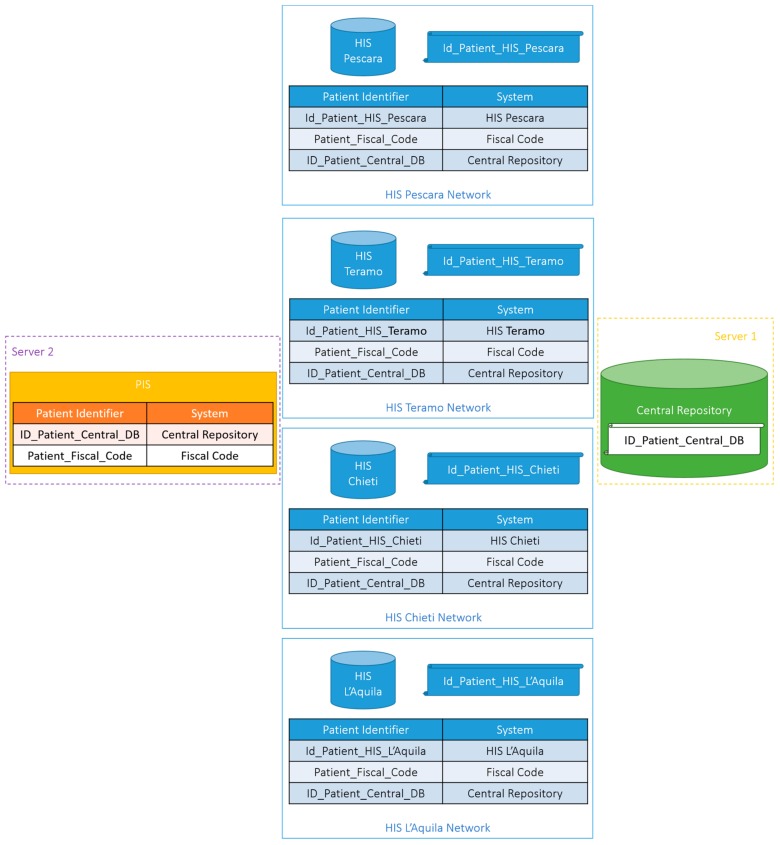
Patient’s identifier management to correctly pseudo-anonymize data.

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
