# Peer review of "Healthcare Associated Infections: An Interoperable Infrastructure for Multidrug Resistant Organism Surveillance"

_ijerph, 2020, doi:10.3390/ijerph17020465_

Round 1

Reviewer 1 Report

The paper describes the technical development of an interoperable infrastructure for multidrug-resistant organisms (MDROs) surveillance for the Abruzzo region in Italy. Conformance to semantic and syntactic interoperability standards, as well as service-oriented architectures and specifications, is described in detail.

In general, the paper uses a very technical language. Therefore, some of the sections are quite difficult to follow, even for an expert. 

The main comment on the manuscript is that the discussion section should be improved. The discussion section starts describing the reasons why the solutions have not been deployed in the Abruzzo region. The discussion, instead, should mainly describe first how the results responded to a research question (explicitly or implicitly ) raised in the introduction section. Then, the innovation of the solution and relevance compared to the state of the art solutions should be described. In general, summarizing the essential findings of the study in the first paragraph is generally recommended.

Despite the fact that the system has not been deployed yet, I recommend that the value of the results is demonstrated in some way. Does it exist a pilot evaluation of the project?. An alternative is a technical evaluation or the solution. For example, would you be able to discuss the properties of the system architecture: e.g., scalability, reusability, flexibility compared to existing solutions?. A question to be solved is why it makes sense to build such a robust architecture? Provide a usability evaluation is also an option if a pilot project doesn’t exist.

This reference explains how to report studies in health informatics:

Other comments:

The English style should be revised. There are very long paragraphs that make it difficult to understand the main idea of a section.

Page 2, line 50. 

ECDC- Explain this abbreviation.

Page 2. Lines 66-72.

“Each enlisted laboratory 66 annually carries out a manual extraction of all microbiological tests carried out only on samples of 67 interest for the GLASS and EAST networks concerning antibiograms (only some antibiotics) related 68 to the 8 pathogens mentioned above and sends them to the national network without adopting either 69 format standards (usually through excel files) or semantic (local terminologies are adopted). ”

It’s not clear whether a semantic and syntactic standard is adopted or not. Revise the long paragraph.

Page 3, line 95.

RPP- Explain this abbreviation.

Pag. 3, Line 101.

Explain why “pseudo-anonymized” data and not wholly anonymized.

Explain also which European data protection and privacy policies and standards followed.

Page 3. Line 106-111. Check text format.

Page 3. Line 116.

Describe in more detail the caption in Figure 1.

Pag. 5. Lines 185, 206.

The List should be comma-separated.

In-Page 5, Please explain how the semantic harmonization process was, especially considering the high number of vocabularies and terminologies. Explain how the HTS service was adapted and implemented.

In section “2.4. Connection with the HIS ”, explains some technical terms or applications?: Areas, Engineering, Thema. The part is difficult to understand.

Discuss why a robust and secure cloud-based data infrastructure is not deployed. Instead, physical servers were used. Is there any backup system?

Reviewer 2 Report

The document aims a very critical area and it seems to present tools for the problems caused by MDROs. It is relevant because it presents a study that is already being implemented in a region of Italy and could present a case study and a guidance for other regions. Figure 2 could be improved as the contrast between colour and white letters is sometimes hard to read. Also the pseudo-anonymization rises some doubts, but not in a bad sense. For one side anonymizing protects privacy on the other side tracing back to some details of the person, such as local reference can help in tracing the habits and even clinical practices. So hopefully the solution is versatile enough to sustain such compromises that, in some sense, could be dynamically managed by ethical boards.
